# Cell-Type-Specific Complement Profiling in the ABCA4^−/−^ Mouse Model of Stargardt Disease

**DOI:** 10.3390/ijms21228468

**Published:** 2020-11-11

**Authors:** Yassin Jabri, Josef Biber, Nundehui Diaz-Lezama, Antje Grosche, Diana Pauly

**Affiliations:** 1Department of Experimental Ophthalmology, Eye Clinic, University Hospital Regensburg, D-93053 Regensburg, Germany; yassin.jabri@ukr.de; 2Department of Physiological Genomics, Biomedical Center, Ludwig Maximilians University Munich, D-82152 Planegg-Martinsried, Germany; Josef.biber@bmc.med.lmu.de (J.B.); Nundehui.Diaz-Lezama@bmc.med.lmu.de (N.D.-L.); 3Experimental Ophthalmology, Philipps-University Marburg, D-35043 Marburg, Germany; 4Department of Ophthalmology, University Hospital Regensburg, D-93053 Regensburg, Germany

**Keywords:** Stargardt macular degeneration, ABCA4, cell-type-specific complement expression, C3, CFI

## Abstract

Stargardt macular degeneration is an inherited retinal disease caused by mutations in the ATP-binding cassette subfamily A member 4 (ABCA4) gene. Here, we characterized the complement expression profile in ABCA4^−/−^ retinae and aligned these findings with morphological markers of retinal degeneration. We found an enhanced retinal pigment epithelium (RPE) autofluorescence, cell loss in the inner retina of ABCA4^−/−^ mice and demonstrated age-related differences in complement expression in various retinal cell types irrespective of the genotype. However, 24-week-old ABCA4^−/−^ mice expressed more *c3* in the RPE and fewer *cfi* transcripts in the microglia compared to controls. At the protein level, the decrease of complement inhibitors (complement factor I, CFI) in retinae, as well as an increased C3b/C3 ratio in the RPE/choroid and retinae of ABCA4^−/−^, mice was confirmed. We showed a corresponding increase of the C3d/C3 ratio in the serum of ABCA4^−/−^ mice, while no changes were observed for CFI. Our findings suggest an overactive complement cascade in the ABCA4^−/−^ retinae that possibly contributes to pathological alterations, including microglial activation and neurodegeneration. Overall, this underpins the importance of well-balanced complement homeostasis to maintain retinal integrity.

## 1. Introduction

Stargardt disease is an inherited macular dystrophy that affects children and young adults. It is characterized by autofluorescent lipofuscin deposition in the retinal pigment epithelium (RPE) and a progressive photoreceptor degeneration [1,2]. Stargardt macular degeneration is caused by mutations involving the ATP-binding cassette subfamily A member 4 (ABCA4) [3]. The ABCA4 transporter is involved in the visual cycle through the clearance of retinoid substrate, thereby avoiding the accumulation of lipofuscin in the RPE derived from phagocytosed photoreceptors [1].

The abnormal accumulation of lipofuscin in the RPE cells causes oxidative stress and complement system activation, leading to progressive macular degeneration and central vision loss [4,5,6,7,8]. The exact process of complement activation in this inherited retinal degenerative disease remains unclear. The complement system is part of the innate immune system and consists of more than 40 proteins that act in a cascade-like manner, modulating the tissue immune homeostasis, e.g., via the opsonization and elimination of foreign microbes as well as damaged cells or by initiating inflammatory responses by the generation of anaphylatoxins [9,10]. Dysregulation of the complement system can lead to uncontrolled inflammation and retinal degeneration. Diseases such as age-related macular degeneration and diabetic retinopathy have shown evidence of altered complement homeostasis within the retina [11,12,13,14].

Previous studies have discovered disturbed complement homeostasis in albino and pigmented ABCA4^−/−^ mice, which included the accumulation of C3 breakdown products together with a reduction of the complement inhibitor factor H at the level of the RPE (Figure 1A) [5,15]. Furthermore, an increase of oxidative stress markers has been described, which led to complement activation in early disease stages [5]. Such changes were followed by a loss of photoreceptors and, thus, a thinning of the outer nuclear layer. They were also accompanied by an enhanced autofluorescence of the RPE and a thickening of the Bruch’s membrane (Figure 1A) [4,5,6,16,17,18].

Here, we describe early changes in C3 blood circulation levels and retinal deposition in the ABCA4^−/−^ mice simultaneously with enhanced autofluorescence in the RPE. This was followed by later accumulation of C3 cleavage products in the RPE accompanied by lower levels of the complement inhibitor complement inhibitors (CFI) in the ABCA4^−/−^ retina. Alterations of the complement homeostasis that appeared during aging seemed to be associated with an increase in numbers as well as signs of moderate activation of microglia and thinning of the ganglion cell layer (GCL). Based on our data, we suggest that rebalancing the complement activity could avoid microglial activation and part of the damage caused during aging and hereditary retinal degeneration.

## 2. Results

### 2.1. Retinal Phenotype is Changed in Albino ABCA4^−/−^ Compared to Wild-Type Mice

To profile retinal changes in the albino mouse model for slow hereditary retinal degeneration lacking the functional ABCA4 transporter (ABCA4^−/−^ mice; Figure 1A), we quantified DAPI-positive cell nuclei in the GCL, inner nuclear layer (INL) and outer nuclear layer (ONL) of mice aged 16‒40 weeks (Figure 1B) and compared these numbers with previously obtained data from wild-type albino mice [19]. The ABCA4^−/−^ mice showed a reduced number of cell nuclei in the GCL compared to wild-type controls starting at the age of 24 weeks. Of note, in addition to retinal ganglion cells, the GCL comprises displaced amacrine cells (about 50% of all GCL cells) and few microglia. The INL was not altered, while the number of the cell nuclei within the ONL was reduced in ABCA4^−/−^ and wild-type mice at the ages of 32 and 40 weeks. This consecutive loss of photoreceptors was accompanied by a mild Müller cell gliosis detected in mice of both genotypes as early as eight weeks of age and onwards (Appendix A). A major consequence of the loss in *abca4* expression is the accumulation of lipofuscin deposits in the RPE and increased autofluorescence thereof (Figure 1C). To correlate retinal complement activation and lipofuscin accumulation, we determined changes in autofluorescence intensity in mice of increasing age (Figure 1D). Indeed, we found a significantly enhanced lipofuscin deposition in the RPE of ABCA4^−/−^ mice older than 24 weeks compared to wild-type controls. However, the RPE monolayer was still intact in both genotypes, as indicated by continuous staining of tight junction protein ZO-1, the cell density and the ratio of cells carrying one, two or more nuclei (Figure 1C and Appendix A).

### 2.2. Complement Expression Changed in the Different Retinal Cell Types Over Time

Performing cell-type-specific analysis of the complement component expression shed light on the local complement homeostasis in the retina of 8‒24-week-old ABCA4^−/−^ mice. Müller cells, microglia, vascular cells and retinal neurons were purified by immunomagnetic separation, and RPE/choroid scratch samples were collected from respective mouse eyes. Successful cell enrichment was validated by the detection of marker genes of respective cell populations. Note that even though astrocytes were not positively selected, we could demonstrate that *gfap* (a marker for astrocytes and reactive Müller cells) is highly enriched in the Müller cell fraction [20]). Given that the number of astrocytes (0.1% of the retinal cell population) per retina is very low compared to Müller cells (3% of the retinal cell population), we termed this macroglial fraction the Müller cell fraction. Moreover, we would like to mention that the neuronal fraction, though not positively selected for a specific neuronal marker, which is hard to do since there is no pan-neuronal marker suitable for immunomagnetic selection, can be considered a rather pure neuronal cell population since it is devoid of microglial (*itgam*, *aif1*), vascular (*icam*, *pecam1*) or macroglial markers (*glul*, *gfap*), as demonstrated in Appendix A. Importantly, marker gene expression was comparable across all tested ages and was independent from the genotype (Appendix A). Only *itgam*, a marker for microglia and macrophages, showed a slightly higher expression in microglia isolated from ABCA4^−/−^ compared to wild-type controls at all investigated time points. 

The complement expression (*c1s*, *c3*, *cfb*, *cfp*, *cfh* and *cfi*) of each cell type was measured by qRT-PCR and compared with the expression levels of wild-type albino mice measured in our previous study [19]. Significant age-dependent changes in complement expression were observed in wild-type albino and ABCA4^−/−^ mice for most analyzed transcripts (Figure 2). Complement expression analysis of *c1s* showed a distinct age-dependent upregulation in Müller cells, vascular cells, neurons and the RPE irrespective of the genotype. In both mouse strains, mRNA for complement components *c3* and *cfb* was increased in the microglia, vascular cells and RPE. Similarly, *cfp* transcripts were elevated in these cell types and Müller cells during aging independent from the genotype. In Müller cells and neurons, the expression of the inhibitory factor *cfh* decreased, while this seemed to be counterbalanced by a strong upregulation in the RPE in ABCA4^−/−^ mice, confirming our previous results in wild-type albino mice [19]. *Cfi*, a cofactor of CFH, was downregulated at the RNA level with aging in microglia, while it was expressed at increasing levels in the RPE. 

Differences in cell-type-specific complement expression for ABCA4^−/−^ mice compared to wild-type albino mice was found at the age of 24 weeks. ABCA4^−/−^ mice upregulated transcripts of complement-activating *c3* in RPE cells compared to wild-type albino mice (Figure 2). Additionally, we detected a downregulation of the complement-inhibiting *cfi* mRNA in aged ABCA4^−/−^ microglia (a trend that was also observed in Müller cells) compared to microglia from wild-type controls (Figure 2). 

These changes in the local complement expression pattern toward proinflammation in ABCA4^−/−^ mice at 24 weeks of age compared to wild-type mice were present at a time point where retinal cell loss was first detected in the GCL and, thus, occurred concomitantly (Figure 1B). 

### 2.3. Increase of C3 Cleavage Products in ABCA4^−/−^ Mice Compared to Wild-Type Mice

Based on the increased *c3* mRNA expression in ABCA4^−/−^ mice, we investigated the protein level of complement component C3 in the retina, RPE/choroid and serum by Western blot analysis (Figure 3A–C, Appendix A) and immunostaining (Figure 3D) using an anti-C3 α-chain antibody specific for the C3d region (His1002-Arg1303). We identified distinct C3 cleavage products, mainly C3α, C3b, iC3b, C3dg and C3d, under reduced conditions in the retina and RPE/choroid (Figure 3A, Figure 4B and Appendix A). In contrast, we detected only the full-length C3α chain and the C3d fragment in the serum (Figure 3C).

The ratios C3b/C3 and C3d/C3 reflect complement activation and the ratio C3b/iC3b gives a better insight into the CFI-dependent conversion from the active C3b to the inactive iC3b. Measuring these ratios gives us a better understanding of complement activation/inactivation than only quantifying total C3. 

We found that the C3b/C3 and C3d/C3 ratios were increased in retinae from 16-week-old albino ABCA4^−/−^ mice compared to wild-type mice (Figure 3A). For the RPE/choroid, we detected an increase of the C3b/C3 ratio in ABCA4^−/−^ mice at 44 weeks of age and a correlated increased C3b/iC3b ratio indicating an activated complement system, which was not counterbalanced by the degradation of C3b into inactive iC3b by CFI.

Apart from local complement changes, we found that the systemic C3d/C3 ratio was increased at 16 weeks of age in ABCA4^−/−^ mice compared to wild-type mice (Figure 3C), which correlated with the C3d/C3 ratio in the retina.

Immunostaining using the anti-C3 antibody targeting the C3d domain of C3 showed higher levels of C3 fragments located on the RPE of albino ABCA4^−/−^ mice compared to controls at 40 weeks of age, which was in line with our Western blot results (Figure 3B,D). Overall, the accumulation of C3 cleavage products in the retina and RPE/choroid of ABCA4^−/−^ mice compared to wild-type mice indicated an activated complement system on protein level in the albino ABCA4^−/−^ mice.

### 2.4. Decrease of CFI Levels in the Retina of ABCA4^−/−^ Mice

A decreased mRNA expression of *cfi* in the microglia cells of ABCA4^−/−^ mice (Figure 2) and an enhanced C3b/iC3b ratio in the RPE/choroid (Figure 3B) suggested a lower activity of the complement regulator CFI. Accordingly, we next validated the CFI expression on protein levels via Western blot and immunostaining in the retinae and serum (Figure 4 and Appendix A). A distinct signal for the full-length CFI (~88 kDa) was detected in the serum samples [21], but only a weak band could be found in the retinal tissue preparations (Figure 4A). We identified a double band for the CFI heavy chain in retinal and serum samples at ~55 kDa, but the corresponding putative double band for the light chain below 35 kDa was smaller than what was theoretically expected (Figure 4A). The intensity of the CFI heavy chain double band was slightly higher at 16 weeks and significantly reduced at 44 weeks of age in the retina of ABCA4^−/−^ compared to wild-type mice (Figure 4B). Similar results were found for the putative CFI light chain (Appendix A). 

Full-length CFI and the CFI heavy chain were also detected in the serum, but we did not find any differences in protein levels between ABCA4^−/−^ and wild-type mice (Figure 4B). In line with the Western blot experiments, lower CFI levels could be determined by immunofluorescent staining in the 40-week-old wild-type and ABCA4^−/−^ retinae, especially at the level of the ganglion cell/nerve fiber layer (Figure 4C).

### 2.5. Signs of Mild Microglial Activation in Aging ABCA4^−/−^ Mice

Increased mRNA expression of complement activators (*c3*, *cfb*, *cfp*), altered protein cleavage products (C3) and decreased complement inhibitor activity (CFI) might be associated with modified immune homeostasis in ABCA4^−/−^ that could be reflected by microglial activation, given that they are the retinal cell type with the highest expression of complement receptors [19,22,23]. Supporting this assumption, the number of detected microglia increased with aging in both mouse strains, which displayed signs of mild activation such as shorter processes as previously reported [19]. Therefore, they occupied a smaller retinal area in 32‒40-week-old mice compared to 24-week-old mice (Figure 5). Interestingly, the number of microglia measured in the inner retinal layers of ABCA4^−/−^ compared to wild-type mice at 32‒40 weeks of age was significantly increased (Figure 5B), which is in line with an enhanced loss of cells in the ganglion cell layer compared to their wild-type counterparts (Figure 1B).

## 3. Discussion

Contribution of an overactive complement system to retinal degeneration along with age-related macular degeneration or Stargardt’s disease have been discussed in the literature [7,8,24,25,26,27]. Studying the albino ABCA4^−/−^ mice as a model for the inherited retinal dystrophy Stargardt disease type [1,4,5,15,16], we described hallmarks of the disease, including cell loss in the GCL [28,29,30], accompanied by a significantly enhanced autofluorescence of the RPE (Figure 1), which was in line with previous reports using comparable ex vivo detection techniques for autofluorescence [4,6].

The observed retinal changes correlated with a distinct complement activation pattern (Figure 3, Appendix A). Importantly, we set out to further validate putative changes in complement activity described by others [5,31]. In this study, we tried to identify the source of these changes in complement activation by defining the contribution of different retinal cell types to the local complement homeostasis and by investigating if systemic complement activity contributed to disease progression in albino ABCA4^−/−^ mice.

Overall, we found mainly age-related retinal complement expression changes consistent in both genotypes. However, a lack of ABCA4 was associated with an increase of *c3* mRNA expression primarily in the RPE and a decrease of *cfi* mRNA in microglia (Figure 2). These data suggested a contribution of retinal cell transcripts to the local complement homeostasis in ABCA4^−/−^ mice. 

Based on our mRNA results, we compared changes in C3 protein level and its activation products in eye tissue and serum samples of ABCA4^−/−^ and wild-type mice. The C3 alpha chain is a 115 kDa protein subunit cleaved by proteases into active (C3b) and inactive (C3d, C3dg) fragments. Interestingly, we identified two major C3 fragments in the mouse serum corresponding either to nonactive C3 or inactivated C3d (Figure 3C), suggesting a well-controlled, systemic complement environment. In contrast, C3 analysis in the retinal tissue uncovered a rather activated status in the retina and RPE/choroid detecting the larger C3 fragments, C3b and iC3b (Figure 3A–C) early in the disease progression. These large, activated C3 fragments were tissue-specific and not detected in the serum. However, a concomitantly increase of systemic inactive C3d fragments correlated with accumulated C3d fragments in the retina at 16 weeks of age in ABCA4^−/−^ mice. This could indicate a relationship between systemic and local complement systems for these small, inactive C3d fragments, but not for the large active C3b fragments. According to our data of an early C3d accumulation in the serum and retina, one could speculate that a temporarily reduced integrity of the blood–retina–barrier (formed by endothelial cells of intraretinal vessels and by the RPE) enabled the influx of systemic C3d into the retina thereby causing increased C3d cleavage product deposition in the retina.

Indeed, it is well known that the C3 cleavage differs in tissue, cell lines and fluids, as well as in healthy and disease conditions [32,33,34,35]. Comparing C3 fragment levels in ABCA4^−/−^ mice to wild-type mice, we identified the RPE as the cell population driving active C3b accumulation in the eye later in disease progression (Figure 3B,D), which did not correlate with systemic changes at 44 weeks of age. In line with our findings, others describe an enhanced deposition of C3 cleavage products at the level of the RPE in mice deficient for ABCA4 using Western blot and immunocytochemistry [5,31] but lacking any information regarding the systemic C3 status. 

This tissue-specific differential pattern of C3 fragments in the murine retina compared to the blood points at a local, retinal regulation of complement activation that might be partially linked to changes of the complement activity in the systemic circulation. Therefore, we proposed that C3b deposition is related to a local tissue response, including loss of cells in the ganglion cell layer and increased microglia cell numbers indicative of neuroinflammatory processes.

Complement component C3 is cleaved in its inactive forms (C3d, C3dg, etc.) by the complement protease CFI and its cofactor CFH that act as complement inhibitors [36,37,38]. Along with the increased *c3* expression, *cfi* mRNA expression was decreased in 24-week-old ABCA4^−/−^ mice. This suggested that dysregulated levels of a complement inhibitor could result in uncontrolled C3 activation and enhanced generation of activated C3b cleavage products in the retinal microenvironment [21,24]. The role of CFI has not yet been extensively studied in the ABCA4^−/−^ model. Lower levels of distinct human CFI variants in patients’ serum were discussed to be associated with AMD [39]. We assumed that the dysregulation of *cfi* expression could be a reason for disease progression in the ABCA4^−/−^ model. In support of this hypothesis, we confirmed lower CFI protein levels in the retina of 40-week-old ABCA4^−/−^ compared to wild-type mice (Figure 4), as well as a constantly increased C3b fragment deposition in ABCA4^−/−^ animals (Figure 3). In line with these results. Rose et al. showed that, in a CFI-deficient mouse, C3 circulates as active C3b with no evidence of other cleavage products (iC3b, C3dg, C3d) [21]. This finding may explain our detection of a higher C3b ratio in the ABCA4^−/−^ mice during degeneration. 

The expression of the second important complement inhibitor *cfh* has recently been shown to be decreased threefold in the eyecup (RPE/choroid/sclera) of four-week-old ABCA4^−/−^ mice [5,31]. Those findings could not be confirmed in our study, which might be explained by differences regarding the age of mice investigated (in this study 8‒24-week-old mice) and the sampling of tissues (in this study, RPE/choroid in contrast to whole eyecup preparations, including scleral tissue used by others). The sclera contains mainly collagen fibers and stroma cells, especially in the outer episcleral layer (including fibroblasts, endothelial cells, melanocytes and infiltrating immune cells) [40]. Krausgruber et al. recently described the three major structural cell types—endothelium, epithelium and fibroblasts—are key immune regulators and express complement genes in various tissues [41]. Voigt et al. have shown, via single-cell RNA sequencing, that *cfh* was exceptionally highly expressed in the fibroblasts in human RPE/choroid samples [42]. This may explain why we did not see an increase of *cfh* in our samples devoid of fibroblast-rich scleral tissue. RPE/choroid and sclera might respond differently to the pathology [42]. We found a low expression of *cfh* transcripts at eight weeks of age in both strains, followed by an upregulation starting from 16 weeks of age, especially in the RPE. On the other hand, there was a trend toward *cfh* downregulation in ABCA4-deficient neurons (Figure 2). This could imply an overall reduction of CFH protein levels in the retina during aging, given that neurons are the retinal cell population expressing the highest *cfh* mRNA levels [19]. Since we focused on significant genotype-related changes that could not be demonstrated for *cfh* mRNA levels, the latter were not further investigated.

In line with an enhanced complement activity in the course of aging, and even more so in the absence of ABCA4, an increase in the number of retinal microglia was found during aging in both mouse strains and was more prominent in ABCA4^−/−^ retinae (Figure 5). Importantly, these changes regarding complement expression and microglial activation were detected in the ABCA4^−/−^ animals before significant photoreceptor cell loss was observed (Figure 1) [43]. Our data correlate with those of others, describing an enhanced microglia activation and the detection of inflammation markers in the retina of ABCA4^−/−^ mice [5,44]. We suggest that a close interplay of changes in the cell-type-specific retinal complement component expression, as well as complement activation product deposition during hereditary retinal degeneration in ABCA4^−/−^ mice and activation of microglia cells, may possibly contribute to slow photoreceptor degeneration.

We conclude that the observed loss of cells in the inner retina, specifically in ABCA4^−/−^ mice, could be due to the dysregulated complement activity in this microenvironment. It has recently been shown that thinning of the INL, the nerve fiber layer, as well as the inner retina, was prominent in different complement knockout mouse models [45]. Such observations suggest that thinning of the inner retina might be a key contributor to malfunction of a complement dysregulated retina or vice versa dysfunction of the complement might affect the integrity of the inner retina.

In sum, activation of the local complement cascade and an interplay with altered systemic complement homeostasis could be a response to an unbalanced, retinal microenvironment due to aging or pathological events in the course of retinal degeneration. Our results imply that rebalancing the local complement activity in inherited retinal diseases may help to prolong neuron survival and avoid potential secondary cell loss, as seen in the GCL of the ABCA4^−/−^ mice. Moreover, this underlines the importance of anti-complement clinical trials in patients with Stargardt disease that in addition to photoreceptor degeneration also suffer from ganglion cell degeneration (such as, e.g., NCT03364153). Furthermore, our cell-type-specific transcription analysis opens a new avenue to identify key genes in retinal degenerative diseases, including those related to aging. 

## 4. Materials and Methods

### 4.1. Animals

Experiments were conducted with 8-, 16-, 24-, 32-, 36-, 40- and 44-week-old male and female mice. Albino ABCA4^−/−^ mice on a BALB/c background were kindly provided by T. Krohne (University of Bonn, Bonn, Germany). To enable the use of littermates, these homozygous ABCA4^−/−^ mice were crossbred with BALB/cJRj mice (Jackson Laboratories). For initial qRT-PCR data collection, 8‒24-week-old BALB/cJRj wild-type mice were used. Data have already been partially published in a previous study addressing age-related complement expression changes [19]. To stick to the rules of the 3Rs, results are included here, such as wild-type controls for the qRT-PCR dataset. All other experiments were performed comparing littermates derived from the ABCA4^−/−^; BALB/cJRj crossbreeding. Mice were housed in a 12 h light/dark cycle with 60‒70 lux inside and 500 lux outside the cages. All experiments were done in accordance with the European Community Council Directive 2010/63/EU and the ARVO Statement for the Use of Animals in Ophthalmic and Vision Research. All animals enrolled in this study were reported to the local authorities (BY_UR_OE_PaulAuge_2016) according to local regulations. Genotyping of the mice was performed with the KAPA Mouse Genotyping Hot Start Kit (Merck, Darmstadt, Germany) using *abca4* primers (Table 1). To ensure that the observed findings of the present study are due to the lack of ABCA4 deficiency, we genotyped the mice for *rd1*, *rd8*, *rd10* and *rd12* (for primers, see Table 1) mutations and verified that no known mutations causing retinal degeneration were present in our mouse breeding. 

### 4.2. Immunohistochemistry of Retina and RPE/Choroid Flat Mounts

Cell nuclei quantification in 10 µm sections of 4% paraformaldehyde (PFA)-fixated, paraffin-embedded eyes was performed with Hoechst33342 (1:1000; #H1399, Thermo Fisher Scientific, Braunschweig, Germany) as previously described [46]. 

Retinal microglia and RPE autofluorescence quantification were evaluated in PFA-fixated, permeabilized and blocked (1% BSA, 5% goat serum, 0.1 M NaPO4, pH 7) retinal or RPE/choroid flat mounts, respectively. RPE/choroid flat mounts were stained with anti-zonula occludens-1 (ZO-1) antibody (1:300, #61-7300, ThermoFisher, Braunschweig, Germany) and secondary anti-rabbit-Cy3 antibody (1:200, #A10520, ThermoFisher, Braunschweig, Germany). Retinal flat mounts were stained with anti-Iba1 antibody (1:400, #019-19741, Wako Chemicals, Neuss, Germany) and secondary anti-rabbit-Cy3 antibody. 

To determine GFAP and complement expression in the retina and RPE, eyes were PFA-fixed, cryoprotected and embedded in Tissue-Tek O.C.T. compound (Cat. No. 4583, #SA62534-10 Sakura, Staufen, Germany). Twenty-micrometer slices were permeabilized, blocked (5% goat serum, 0.1% Tween20, PBS) and incubated with primary antibodies specific for GFAP (1:500, G3893, Sigma, Saint-Louis, MO, USA ), CFI (1:100, A313/6, Quidel, San Diego, CA, USA) or C3d (12 µg/mL, AF2655, R&D Systems, Minneapolis, MN, USA) diluted in a blocking buffer. Sections were labeled with Cy3-conjugated secondary antibodies (1:500, Cat. No. 705-165-147, Dianova, Hamburg, Germany) and DAPI nucleic acid stain (1:1000, Cat. No. 62248c, Invitrogen, Ltd., Paisley, UK). Slides were mounted with Aqua-Poly/Mount (Cat. No. 18606-20, Polysciences, Hirschberg, Germany). All images were taken with a confocal microscope (VisiScope, Visitron Systems, Puchheim, Germany).

### 4.3. Isolation of Retinal Cell Populations by Immunomagnetic Enrichment

Retinal cell types were enriched as described previously [46]. Briefly, retinae were treated with papain (0.2 mg/mL; Roche Molecular Biochemicals) in a Ca^2+^- and Mg^2+^-free extracellular solution (140 mM NaCl, 3 mM KCl, 10 mM HEPES, 11 mM glucose, pH 7.4). After DNase I (200 U/mL) incubation, retinae were triturated in an extracellular solution (1 mM MgCl_2_ and 2 mM CaCl_2_). To purify microglial and vascular cells, the retinal cell suspension was subsequently incubated with CD11b and CD31 microbeads according to the manufacturer’s protocol (Miltenyi Biotec, Bergisch Gladbach, Germany). The respective binding cells were depleted from the retinal suspension using large cell (LS) columns (Miltenyi Biotec) prior to Müller cell enrichment. To purify Müller glia, the cell suspension was incubated in an extracellular solution containing biotinylated hamster anti-CD29 (clone Ha2/5, 0.1 mg/mL, BD Biosciences, Heidelberg, Germany). Cells were washed in the extracellular solution, spun down, resuspended in the presence of antibiotin MicroBeads (1:5; Miltenyi Biotec,) and incubated for 10 min at 4 °C. After washing, CD29+ Müller cells were separated using LS columns according to the manufacturer’s instructions (Miltenyi Biotec). Cells in the flow through the last sorting step were considered a neuronal population as they were depleted from microglia, vascular cells and Müller glia. The retinal pigment epithelium was collected by scratching it out of the eyecup after the retina had carefully been removed and, thus, scratch samples also contained cells from the underlying choroid.

### 4.4. qRT-PCR

Total RNA was isolated from the enriched cell populations using the PureLink RNA Micro Scale Kit (Thermo Fisher Scientific, Schwerte, Germany). A DNase digestion step was included to remove genomic DNA (Roche). We performed RNA integrity validation and quantification using the Agilent RNA 6000 Pico chip analysis according to the manufacturer’s instructions (Agilent Technologies, Waldbronn, Germany). First-strand cDNAs from the total RNA purified from each cell population were synthesized using the RevertAid H Minus First-Strand cDNA Synthesis Kit (Fermentas by Thermo Fisher Scientific, Schwerte, Germany). We designed primers using the Universal ProbeLibrary Assay Design Center (Roche) and measured transcript levels of candidate genes by qRT-PCR using the TaqMan hPSC Scorecard Panel (384-well, ViiA7, Life Technologies, Darmstadt, Germany) according to the company’s guidelines [19]. Expression levels of respective complement genes (for primer sequences, see Table 2) were normalized to the expression of isocitrate dehydrogenase 3 (NAD+) beta (*idh3b*) that we validated in a recent studied to fulfill the criteria of a good housekeeping gene across retinal cell populations [19].

### 4.5. Western Blot 

Retina and scratch RPE/choroid were pooled from two mouse eyes and homogenized in T-PER™ buffer (Pierce Biotechnology, Rockford, IL, USA) supplemented with protease and phosphatase inhibitors (Sigma-Aldrich, Taufkirchen, Germany). The homogenized retina, RPE/choroid or isolated serum were denatured in reducing a Laemmli sample buffer and separated on a 12% SDS-PAGE. The immunoblot was performed as previously described [47]. Detection was performed using a goat polyclonal C3 (1:1000, AF2655, R&D Systems, Minneapolis, MN, USA), a goat anti-CFI antibody (1:200, A313/6, Quidel, San Diego, CA, USA) or a rabbit anti-PDHB antibody (1:2000, Abcam, ab155996, Cambridge, UK), followed by anti-goat (1:10,000, Dianova, Hamburg, Germany) or anti-rabbit (1:10,000, Dianova, Hamburg, Germany) secondary antibodies conjugated to horse-radish peroxidase. Respective primary and secondary antibodies were diluted in a blocking solution. Blots were developed with Western Sure PREMIUM Chemiluminescent Substrate (LI-COR, Bad Homburg, Germany).

### 4.6. Statistics

Statistical analyses were performed using Prism (GraphPad Prism 7.04, San Diego, CA, USA). In most of the experiments in the present study, results from 4 biological replicates were collected to keep to the rules of the three Rs for the sake of animal welfare. Since this low number of input values does not allow an appropriate estimation of a normal Gaussian distribution, significance levels were determined by the nonparametric, two-tailed Mann–Whitney U-test, unless stated otherwise. All data are expressed as mean ± standard error (SEM) unless stated otherwise. Detailed information about specific n-values, implemented statistical tests and coding of significance levels are provided in the respective figure legends.

## Figures and Tables

**Figure 1 ijms-21-08468-f001:**
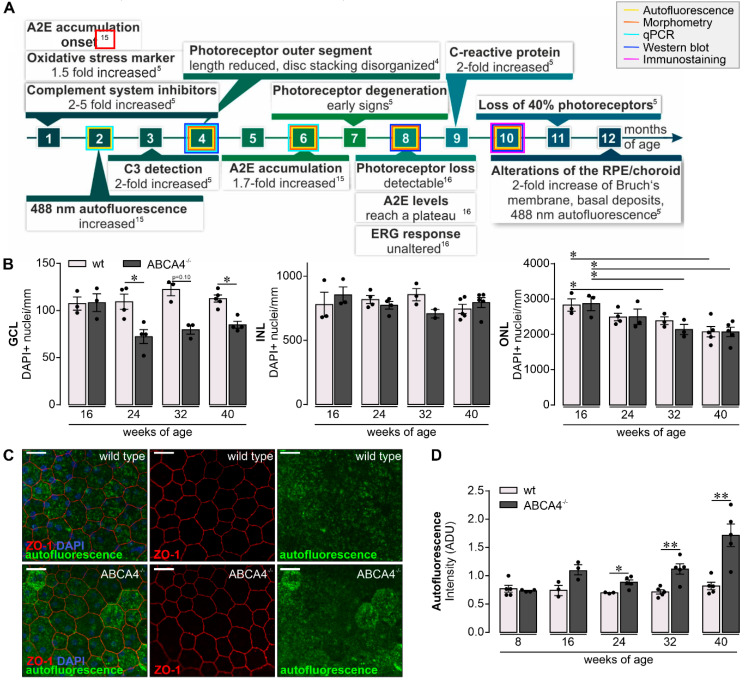
Retinal phenotype in the investigated albino ABCA4^−/−^ strain. (**A**) Subsumed retinal phenotype of albino ABCA4^−/−^ mice based on published studies [4,5,15,16]. ERG, electroretinogramm; ONL, outer nuclear layer; A2E, N-retinylidene-N-retinylethanolamine; C3, complement factor 3. Experiments performed in the course of the present study at different time points are highlighted by frames in respective colors. (**B**) A decrease in the number of DAPI+ cell nuclei in the ganglion cell layer (GCL) is observed in ABCA4^−/−^ mice aged 24, 32 and 40 weeks but not in the albino wild-type retina (data were obtained previously) [19]. Cell numbers in the retinal inner nuclear layer (INL) were stable in aging ABCA4^−/−^ mice. Photoreceptor density, as determined by quantification of DAPI+ cell nuclei in the outer nuclear layer (ONL), was significantly decreased in ABCA4^−/−^ and wild-type mice at 32 weeks of age and older [19]. (**C**) The integrity and autofluorescence of the retinal pigment epithelium (RPE) were investigated in fixed, flat-mounted eyecup preparations after careful removal of overlaying retinae. ZO-1 labeling delineates tight junctions formed by intact RPE cells. Autofluorescence was detected upon excitation with a 488 nm laser and revealed enhanced signals at 40 weeks of age from the RPE of ABCA4^−/−^ mice as compared to the RPE from wild-type littermates. Scale bars, 20 µm. (**D**) Mean gray values for autofluorescence were measured over the whole scan field. (**B**,**D**). Bars represent mean values ± SEM from 2 to 5 animals. * *p* < 0.05, ** *p* < 0.01, Mann–Whitney U-test (two-tailed). ADU, arbitrary digital units.

**Figure 2 ijms-21-08468-f002:**
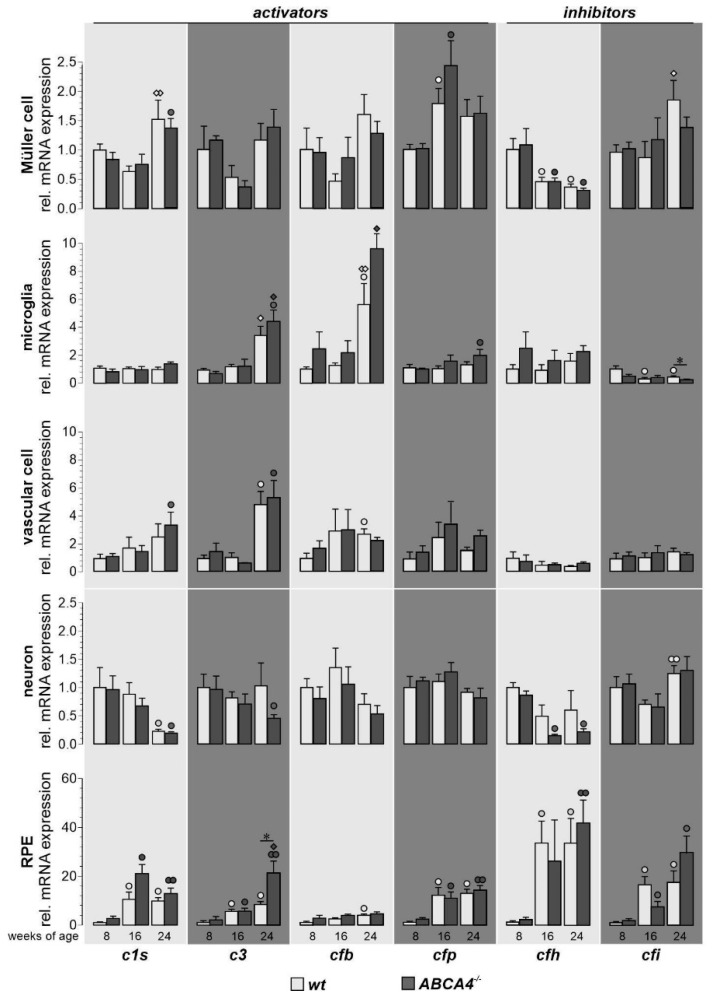
Comparison of complement component expression between retinal cell types of wild-type albino and ABCA4^−/−^ mice. Complement expression analysis of *c1s*, *c3*, *cfb*, *cfp*, *cfh* and *cfi* by qRT-PCR was performed on Müller cells, microglia, vascular cells, neurons and RPE from 8-, 16- and 24-week-old ABCA4^−/−^ mice and compared to previously published wild-type data [19]. We found a significantly enhanced expression of *c3* in RPE cells and decreased expression of *cfi* in microglia cells compared to wild-type controls. Most other age-dependent changes in complement expression were similar in both mouse strains [19]. Bars represent mean values ± SEM of cells purified from four to six animals. Mann–Whitney U-testing was performed on all data (**p* < 0.05. White circle (◦): significant difference compared to the expression level at eight weeks of age in wt; black circle (●): significant difference compared to the expression level at eight weeks of age in ABCA4^−/−^ mice; white diamond (◊): significant difference compared to the expression level at 16 weeks of age in wt animals; black diamond (♦): significant difference compared to the expression level at 16 weeks of age in wt animals. ◦/●/◊/♦:
*p* < 0.05; ●●/◊◊: *p* < 0.01).

**Figure 3 ijms-21-08468-f003:**
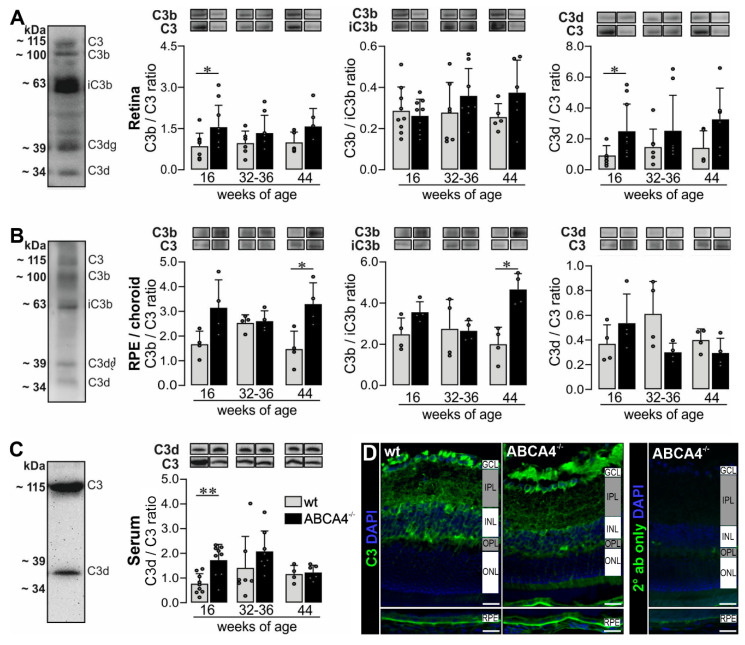
Western blot analysis and immunostaining of complement factor C3 in the RPE/choroid, retina and serum of wild-type albino and ABCA4^−/−^ mice. (**A**,**B**) Western blot analysis under reduced condition using an anti-C3 α-chain antibody showed bands for C3, C3b, iC3b, C3dg and C3d fragments in the (**A**) retina and (**B**) RPE/choroid. (**A**) At 16 weeks of age, we detected a significant increase of C3b to C3 and C3d to C3 ratio in the retina of ABCA4^−/−^ mice compared to wild-type mice. (**B**) The ratio of C3b to C3 and C3b to iC3b was significantly increased in the RPE/choroid of ABCA4^−/−^ mice compared to wild-type mice at 44 weeks of age. (**C**) C3 and C3d fragments were detected in the serum using the C3d-fragment-specific antibody. The C3d to C3 ratio was increased at 16 weeks of age in ABCA4^−/−^ mice. (**D**) Immunofluorescence staining using an anti-C3 (green/gray) antibody showed an increase of C3 fragments in the RPE in albino ABCA4^−/−^ mice compared to wild-type mice. Cell nuclei were counterstained with DAPI (blue). Stainings were performed on sections from at least three mice per genotype and representative images were chosen. As a negative control, sections were incubated with the secondary antibody (2° ab) only. GCL, ganglion cell layer; IPL, inner plexiform layer; INL, inner nuclear layer; OPL, outer plexiform layer, ONL, outer nuclear layer. Scale bars, 20 µm. (**A**–**C**) Bars represent mean values ± SEM from 4 to 10 animals. * *p* < 0.05, ** *p* < 0.01, Mann–Whitney U-test.

**Figure 4 ijms-21-08468-f004:**
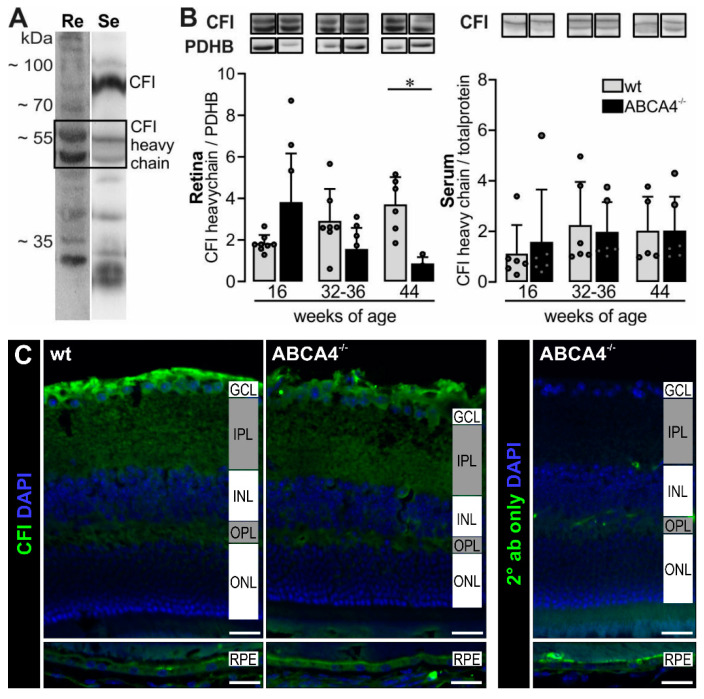
Complement inhibitors (CFI) heavy chain levels decreased in aging ABCA4^−/−^ mice. (**A**) CFI heavy chain was detected in the retina (Re) and serum (Se) using Western blot under reduced condition. (**B**) Photometric quantification of CFI heavy chain showed a significant decrease in ABCA4^−/−^ mice at 44 weeks of age compared to wild-type in the retina. Pyruvate dehydrogenase E1 subunit beta (PDHB, 30–35 kDa) was used to normalize the CFI levels in the retina. For the serum, no differences were found between respective genotypes. (**C**) Immunofluorescence detection of CFI in the cross-sections of mouse eyes. Nuclei were counterstained with DAPI. Stainings were performed on sections from at least three mice per genotype and representative images were chosen. As a negative control, sections were incubated with the secondary antibody (2° ab) only. GCL, ganglion cell layer; IPL, inner plexiform layer; INL, inner nuclear layer; OPL, outer plexiform layer, ONL, outer nuclear layer. Scale bars, 20 µm. (**A**,**B**) Bars represent mean values ± SEM from 4 to 10 animals. * *p* < 0.05, Mann–Whitney U-test.

**Figure 5 ijms-21-08468-f005:**
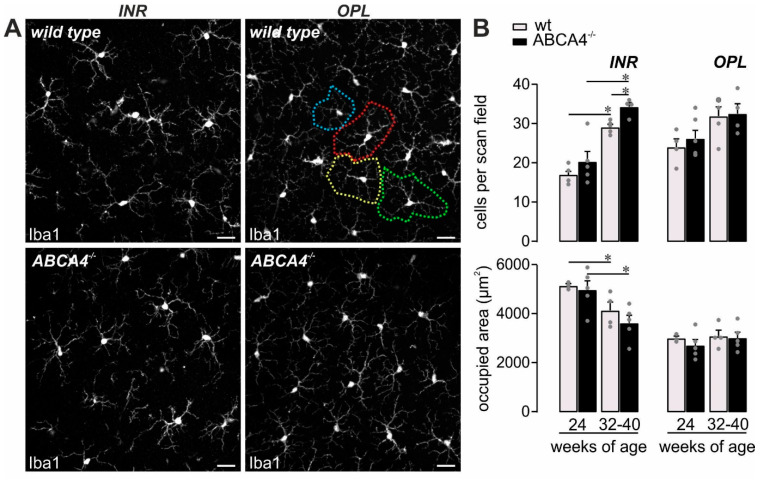
Mild microglia activation in ABCA4^−/−^ mice. (**A**,**B**) Microglia were quantified in the inner retinal layers (INR), such as the ganglion cell layer and inner plexiform layer, and additionally in the outer plexiform layer (OPL) on the basis of Iba1 labeling in mice of the indicated age in ABCA4^−/−^ mice and compared to previously published data of wild-type albino mice [19]. (**A**) The area occupied by the widely branched processes of a single microglia was measured as exemplarily depicted by the dashed circles of different colors for the OPL microglia in a retina from a wild-type animal. Scale bars, 20 µm. (**B**) Bars represent mean values ± SEM from 2‒4 animals. * *p* < 0.05, Mann–Whitney U-test. ADU, arbitrary digital units.

**Table 1 ijms-21-08468-t001:** Primer sequences for mouse genotyping by PCR.

Gene ID	Primer Sequences: *Forward*	Primer Sequences: *Reverse*	Accession Number
*abca4*	5′ aggagaagcaatcaaatcagga 3′	5′ gaagatgctctggatatctctgc 3′ 5′ tgagtaggtgtcattctattctgg 3′	NM_007378.1
*rd1*	5′ tgacaattactccttttccctcagtctg 3′ 5′ tacccacccttcctaattttctcacgc 3′	5′ gtaaacagcaagaggctttattgggaac 3′	AH002075.2
*rd8*	5′ gtgaagacagctacagttctgatc 3′ 5′gcccctgtttgcatggaggaaacttggaagacagctacagttcttctg 3′	5′ gccccatttgcacactgatgac 3′	NM_133239.2
*rd10*	5′ acaaggaacaagggctctga 3′	5′ ccttccactcattgctaggac 3′	NM_008806.2
*rd12*	5′ tgacacctagttttaatattttgatcc 3′	5′cagagcttgaaccccatt 3′	NM_029987.2

**Table 2 ijms-21-08468-t002:** Primer and TaqMan probe combinations for the detection of complement components via qRT-PCR.

Gene ID	Primer Sequences: *Forward*	Primer Sequences: *Reverse*	TaqMan^®^ Probe from Roche	Accession Number
*idh3b*	5′ gctgcggcatctcaatct 3′	5′ ccatgtctcgagtccgtacc 3′	# 67	NM_130884.4
*c1s*	5′ ggtggatacttctgctcctgtc 3′	5′ agggcagtgaacacatctcc 3′	# 69	NM_144938.2
*c3*	5′ accttacctcggcaagtttct 3′	5′ ttgtagagctgctggtcagg 3′	# 76	NM_009778.3
*cfb*	5′ ctcgaacctgcagatccac 3′	5′ tcaaagtcctgcggtcgt 3′	# 112	NM_008198.2
*cfp*	5′ tcttgagtggcagctacagg 3′	5′ cagaccagccacccatct 3′	# 56	NM_008823.4
*cfh*	5′ aaaaaccaaagtgccgagac 3′	5′ ggaggtgatgtctccattgtc 3′	# 25	NM_009888.3
*cfi*	5′ tttctcttggctctccacttg 3′	5′ tgcagtaagcatttctgatcg 3′	# 63	NM_007686.3

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
