# Peer review of "Cell-Type-Specific Complement Profiling in the ABCA4−/− Mouse Model of Stargardt Disease"

_ijms, 2020, doi:10.3390/ijms21228468_

Round 1
Reviewer 1 Report
The paper presented by Jabri et al entitle “Cell type-specific complement profiling in the ABCA4-/- mouse model of Stargartd disease” is a well-structured and results very are interesting for the retinal diseases field. In my opinion the paper is almost ready for publication, only a couple of issues needs to be addressed.
Authors quantified the nuclei stained by DAPI at the ganglion cell layer and in some sentences of the work it can be read “ganglion cell loss” as for example in lines 223 and 286. Authors need to avoid these affirmations since ganglion cells in ganglion cell layer represent only the 50%, in this retinal layer there are other type of cells like displaced amacrine cells and even microglia so the loss of nuclei in this layer could be due to other type of cells.
Regarding the quantification of cells in RGL, can authors explain why there’s not significant loss at 32 weeks, while at 24 and 40 weeks there is statistically significant differences in cells at the RGL (figure 1B).
Regarding to the complement components expression in different retinal cell types. I think authors should consider to change the name to the group authors call neurons or discard this results. Authors do not use any specifically marker to isolate the cluster. Authors say that this group of cells are “neuronal population as it was now depleted from microglia, vascular cells and Müller cells” (line 380). Authors forget that there still glial cells in retina tissue, astrocytes, therefore this group of cells should not be considered only as neurons.
Author Response
- Authors quantified the nuclei stained by DAPI at the ganglion cell layer and in some sentences of the work it can be read “ganglion cell loss” as for example in lines 223 and 286. Authors need to avoid these affirmations since ganglion cells in ganglion cell layer represent only the 50%, in this retinal layer there are other type of cells like displaced amacrine cells and even microglia so the loss of nuclei in this layer could be due to other type of cells.
Thank you, for raising this valid point. We did revise the manuscript accordingly and avoided the use of the term “ganglion cell loss”.
- Regarding the quantification of cells in RGL, can authors explain why there’s not significant loss at 32 weeks, while at 24 and 40 weeks there is statistically significant differences in cells at the RGL (figure 1B).
Unfortunately, we could not include more than 3 individuals per group for 32 weeks old animals, since the mouse colony had to be shut down for reasons we had no influence on. The complete animal facility had to be shut down. Since this mouse strain on the Balb/c background is not commercially available and the original source (colony in the lab of Tim Krohne, Bonn, Germany) was discontinued, we had no chance to add more data points to this experimental group. If a one-tailed Mann-Whitney test would be performed with these data that include a n=3 (assuming that only a reduction of cell number is possible given that retinal neurons do not proliferate), the p-value would be <0.05. Since we try to avoid taking too many assumptions, we however prefer to use two-tailed tests. As pointed out by the reviewer, the ganglion cell layer does also contain microglia that indeed could proliferate. We now added the exact p-value into the figure and discussed this briefly in the results part.
- Regarding to the complement components expression in different retinal cell types. I think authors should consider to change the name to the group authors call neurons or discard this results. Authors do not use any specifically marker to isolate the cluster. Authors say that this group of cells are “neuronal population as it was now depleted from microglia, vascular cells and Müller cells” (line 380). Authors forget that there still glial cells in retina tissue, astrocytes, therefore this group of cells should not be considered only as neurons.
Thank you for this comment. The reviewer is totally right. We do not positively select retinal neurons with our protocol. However, from our recent data we can conclude that e.g. astrocytes definitely do not reside in the fraction we termed “neuronal”. GFAP as one marker for astrocytes is clearly enriched in the Müller cell fraction meaning that astrocytic “contamination” is found in the Müller cell samples (see below in Fig. 1). However, since both belong to the macroglia lineage and given that Müller cells by far outnumber astrocytes, we are confident that our data on complement expression in our Müller cell population really represents what happens in Müller cells. To meet that concern of the reviewer, we now included these explanations also into the manuscript in the methods part and in the results section where the data on transcript analysis are presented/discussed to clarify this for the reader. Given that we positively select for micro- and macroglia as well as vascular cells, we would therefor like to keep the term neuronal population/fraction.
Fig. 1 (see attached pdf-file). qPCR data demonstrating gfap expression in the Müller cell fraction derived from a data set of the study Mages et al., 2019, Journal of Neuroinflammation. They demonstrate that gfap, that typically serves as marker gene for astrocytes and reactive Müller cells, is clearly expressed in the Müller cell fraction, but not in neurons. Retinal cells were purified via the same protocol as used in the present study and were generated from non-injured, healthy eyes of 2 - 4 months old C57Bl/6J mice. Bars represent mean values ± SEM (n = 3–6 mice). Mc, Müller cells; mg, microglia; vc, vascular cells; n, neurons.

Reviewer 2 Report
The present paper presents accurate data on the expression and protein content of complement C3 and CFI in cells of healthy control and ABCA4 knock out mice at different times during disease progression. The experimental procedure has been well presented but there the results might be discussed more clearly. Looking at fig. 1 B it seems that there is a severe loss of ganglion cells in ABCA4 -/- mice did the authors used Tunel staining (for example) to label dying neurons? This point is interesting also looking at photoreceptors where the loss is the same in aging and ABCA4-/-. Also the time course of autofluorescence is strange. The increased in the mRNA expression for activators and inhibitors seems to go more with aging that with disease (with some exception). I remain confused about the role played by complement during degenerative process. CF1 dys-regulation is particularly evident at 44 weeks (one-year-old mice) I wonder whether it is possible to conclude (line 281) that this is “a potent reason for disease progression”. I think the retinal sections in fig.4 are from mice 44 months old the quality of the image is questionable. In any case the alteration of complement homeostasis seems to be more linked to the inner retina remodeling, following photoreceptors malfunction, (microenvironment?), this point might be discussed further. I think the discussion has to be reorganized to offer clear messages and hypothesis.
Author Response
Point-by-point reply
Reviewer 2
- Looking at fig. 1 B it seems that there is a severe loss of ganglion cells in ABCA4 -/- mice did the authors used Tunel staining (for example) to label dying neurons? This point is interesting also looking at photoreceptors where the loss is the same in aging and ABCA4-/-.
We agree with the reviewer that one could add more methods to validate cellular loss. Since we focussed on the role of complement in this already quite well established mouse model, we did not go into too much detail in this respect. Previous publication detected TUNEL positive cells only after light damage of the retina in Abca4-/-Rdh8-/- mice, but not in untreated animals (Maeda et al. 2009, PMID: 19304658; Liao et al. 2020, PMID: 32265302).
We now used remaining cryosections from some of the 40 week old animals and confirmed these findings, detecting no TUNEL-positive cells in untreated ABCA4-/- mice (Fig. 2). As the reviewer pointed out, we would have expected to at least observe TUNEL-positive cells in the ONL in both genotypes, but obviously, cell loss is extremely slow or does not involve apoptosis.
- Also the time course of autofluorescence is strange.
Thank you for this comment. Indeed we detected a late, significant increase of RPE-dependent autofluorescence at 24 weeks of age in the described albino ABCA4-/- mouse model compared to wildtype albino mice which correlates with other findings (Taubitz et al. 2018, PMID: 30038866, Lenis et al. 2017, PMID: 28348233). In contrast, Sparrow et al. and Radu et al. detected enhanced RPE-dependent autofluorescence already at 8 weeks of age in albino ABCA4-/- which was further increased in 32 week old mice (Sparrow et al. 2013, PMID: 23548623; Radu et al. 2011, PMID: 21464132).
These differences may be due to different sensitivities of the used detection systems. In this, the Taubitz and Lenis et al. study RPE autofluorescence was determined by a confocal microscope in ex vivo RPE flat mounts or tissue slides. Others investigated RPE autofluorescence in living mice using a confocal scanning laser ophthalmoscope (Sparrow et al. 2013, PMID: 23548623).
It seems that ex vivo microscopic analysis of RPE, as it was also performed in our study, is less sensitive than in vivo imaging techniques. However, ex vivo studies come with the advantage that further RPE characteristics can be determined e.g. ZO-1 localization (as done in our study) or ultrastructural alterations (as done in Taubitz et al. 2018, PMID: 30038866).
In sum, we think that the autofluorescence data included in our manuscript are in accordance with other studies using comparable methods and present them as additional data set to validate the phenotype in our specific mouse breeding. However, if the reviewer specifies his/her concerns, we would be happy to further discuss details.
- The increased in the mRNA expression for activators and inhibitors seems to go more with aging that with disease (with some exception).
We totally agree with the reviewer that most changes of complement expression seem to be age-related and only very few are clearly associated with the ABCA4 knockout. We tried our best to clearly specify which changes are most likely due to aging and which are indeed caused by the knockout in the discussion of our results. We revised the discussion carefully to clearly discriminate between these two effects.
- I remain confused about the role played by complement during degenerative process. CFI dysregulation is particularly evident at 44 weeks (one-year-old mice) I wonder whether it is possible to conclude (line 281) that this is “a potent reason for disease progression”.
Local mRNA expression of c3 and cfi is significantly changed in ABCA4-/- mice compared to wild type mice at 24 weeks of age. C3 mRNA is increased and cfi transcripts are decreased. In consequence, we detected much more activated C3 in the retina of ABCA4-/- deficient mice, than in wildtype mice. This correlated with the time point when we detected a decrease in cell numbers in the GCL as well as increase of autofluorescence in the RPE. These data suggested a relationship between retinal degeneration and complement components ABCA4-/- mice.
Indeed, reduced cfi mRNA levels did not lead directly to decreased CFI protein levels in the retina detected at 24 weeks, rather later at 44 weeks. However, we cannot exclude an influx of systemic CFI into the retina by temporarily reduced integrity of the blood-retina-barrier (as stated for C3 in line 297 of the discussion), which could normalized reduced levels of locally secreted CFI and balance the local complement components early in the degeneration process.
In respect to the reviewer comment, we revised line 321 in the manuscript.
- I think the retinal sections in fig.4 are from mice 44 months old the quality of the image is questionable.
We thank the reviewer for that comment even though we are not totally sure what the reviewer refers to. Does it address the overall resolution of the micrographs? Maybe this was a problem with image compression. We now uploaded also Tif-images to avoid such kind of problems. If it addresses the quality of the tissue section, we would like to state that CFI staining per se is rather all over the place in the retina with highest staining intensities at Müller cell endfeet. Müller cell endfeet in knockout mice at times look less well preserved. This may explain why also CFI staining is less intense. But it is hard to conclude from immunolabeling data what is cause and what the effect so we decided only to describe what we observe.
- In any case the alteration of complement homeostasis seems to be more linked to the inner retina remodeling, following photoreceptors malfunction, (microenvironment?), this point might be discussed further.
Thank you for pointing out this very important aspect. We added some more thoughts into the discussion to appreciate the potential impact of complement on inner retinal reorganisation/degeneration in the ABCA4-/- mice (see line 360-365 in the revised discussion).
- I think the discussion has to be reorganized to offer clear messages and hypothesis.
We worked hard on the discussion to restructure it and thereby provide clear messages and hypothesis. Changes are tracked in the revised manuscript.

Round 2
Reviewer 2 Report
the authors clearly answered all the questions raised by the referee. The paper was nicely improved.